# Detection of Antibodies against Endemic and SARS-CoV-2 Coronaviruses with Short Peptide Epitopes

**DOI:** 10.3390/vaccines11091403

**Published:** 2023-08-23

**Authors:** Michael Szardenings, Nicolas Delaroque, Karolin Kern, Lisbeth Ramirez-Caballero, Marcus Puder, Eva Ehrentreich-Förster, Joachim Beige, Sebastian Zürner, Georg Popp, Johannes Wolf, Stephan Borte

**Affiliations:** 1Fraunhofer Institute for Cell Therapy and Immunology (IZI), Perlickstr. 1, 04103 Leipzig, Germany; nicolas.delaroque@izi.fraunhofer.de (N.D.); karolin.kern@izi.fraunhofer.de (K.K.); zuerner@wifa.uni-leipzig.de (S.Z.); georg.popp@izi.fraunhofer.de (G.P.); 2epitopic GmbH, Deutscher Platz 5e, 04103 Leipzig, Germany; marcus.puder@epitopic.com; 3Fraunhofer Institute for Cell Therapy and Immunology, Branch Bioanalytics and Bioprocesses IZI-BB, Am Mühlenberg 13, 14476 Potsdam, Germany; eva.ehrentreich@izi-bb.fraunhofer.de; 4Martin-Luther-University Halle/Wittenberg, Medical Clinic 2, 06112 Halle, Germany; joachim.beige@kfh.de; 5WINF/Informationsmanagement, University Leipzig, Grimmaische Straße 12, 04109 Leipzig, Germany; 6Department of Laboratory Medicine, Hospital St. Georg, Delitzscher Strasse 141, 04129 Leipzig, Germany; johannes.wolf@sanktgeorg.de (J.W.); stephan.borte@sanktgeorg.de (S.B.); 7ImmunoDeficiencyCenter Leipzig (IDCL), Jeffrey Modell Diagnostic and Research Center for Primary Immunodeficiency Diseases, Hospital St. Georg, Delitzscher Strasse 141, 04129 Leipzig, Germany

**Keywords:** coronavirus, COVID-19, epitopes, cross-reactivity, serology, peptide array, array image processing, biomarkers

## Abstract

(1) Background: Coronavirus proteins are quite conserved amongst endemic strains (eCoV) and SARS-CoV-2. We aimed to evaluate whether peptide epitopes might serve as useful diagnostic biomarkers to stratify previous infections and COVID-19. (2) Methods: Peptide epitopes were identified at an amino acid resolution that applied a novel statistical approach to generate data sets of potential antibody binding peptides. (3) Results: Data sets from more than 120 COVID-19 or eCoV-infected patients, as well as vaccinated persons, have been used to generate data sets that have been used to search in silico for potential epitopes in proteins of SARS-CoV-2 and eCoV. Peptide epitopes were validated with >300 serum samples in synthetic peptide micro arrays and epitopes specific for different viruses, in addition to the identified cross reactive epitopes. (4) Conclusions: Most patients develop antibodies against non-structural proteins, which are useful general markers for recent infections. However, there are differences in the epitope patterns of COVID-19, and eCoV, and the S-protein vaccine, which can only be explained by a high degree of cross-reactivity between the viruses, a pre-existing immune response against some epitopes, and even an alternate processing of the vaccine proteins.

## 1. Introduction

The human antibody repertoire has a valuable memory for immunological encounters that are either ongoing, recent, or, in some cases, those that occurred a long time ago. The B-cell response also varies between patients, generating individual fingerprints of epitope patterns recognized on an antigen. A single drop of blood should contain hundreds or more molecules of each antibody, for at least the most dominant epitopes of the related antigens. To unlock this resource with standard methods and the usually limited amounts of serum available per patient for epitope identification, researchers have to focus on a few selected proteins. The use of peptide arrays is well established, but it is limited to preselected protein sequences, becomes tedious, and requires larger quantities of patient serum when screening for variations in virus strains or, as in the case of coronaviruses, screening for a whole family’s divergent proteins. 

To overcome this problem, a special peptide phage library stringently designed to cover all included sequences with a predictable frequency [1] was applied here for the first time to a large number of sera from infectious disease patients. This allows for a statistical analysis based on peptide fragments instead of full sequences. A single selection of the entire pool of serum antibodies resulted in a small number of sequences bound by antibodies from individual B-cell clones. Because of the nature of the peptide gene library, NGS can generate a database of peptide sequences, which can be searched in vitro for enriched peptide motifs matching any suggested antigen sequence, replacing hundreds or thousands of tests that consume large quantities of serum and require the preparation of the relevant proteins. To search for epitopes of the coronavirus proteome, we applied this approach for the first time to hundreds of sera from COVID-19, respiratory disease, and vaccinated patients. In addition, a special image analysis method is required for rapid and reliable validation of a large number of peptide arrays, overcoming bottlenecks using a semi-manual analysis. By comparing the results from different patients, minimized epitope/mimotope peptides were identified. This can be highly specific to pathogenic strains.

The recent pandemic has pushed general attention towards SARS-CoV-2, a novel coronavirus. The virus first emerged in late 2019, but is not the only pathogenic coronavirus, although it is the most life-threatening and globally spreading strain after SARS-CoV, which vanished shortly after its occurrence. Coronaviruses (CoVs) are a group of highly diverse, enveloped, positive-sense, single-stranded RNA viruses [2]. More than 50 coronaviruses have been discovered and sequenced; in addition to SARS-CoV-2, six human coronaviruses (hCoVs), four seasonal coronaviruses (hCoV-229E, -NL63, -HKU1, and -OC43), and the two most recently discovered viruses, SARS-CoV and MERS-CoV, originating from recent zoonotic events, have been discovered [3,4]. Human pathogenic CoVs (HCoVs), such as HCoV-OC43 and HCoV-229E, are known to cause mild upper respiratory diseases, contributing to 5–30% of the seasonal common cold cases [5,6]. This explains why more than 90% of the global population has antibodies against common cold CoVs [7]. According to several studies, it also seems to be common that two different strains infect a single patient at the same time [8,9,10,11], which would generate a special challenge for the patient’s immune system.

The proteome, proteins, architecture, and structure of the virus particles are similar. Cold-causing coronaviruses (for example, OC43 and 229E strains) are quite similar to SARS-CoV-2 in genome length (within 10%) and gene content, but different from SARS-CoV-2 in sequence (>50% nucleotide identity) [12]. SARS-CoV-2 shows 80% sequence identity with SARS-CoV and 50% identity with MERS-Cov, respectively [13,14]. The genome of SARS-CoV-2 is approximately 29,903 nt and encodes four structural and 16 non-structural proteins (NSPs) [15]. Structural proteins include the spike (S), envelope (E), membrane (M), and nucleocapsid (N). Sixteen NSPs were encoded by the open reading frame 1a (ORF1a) and ORF1b. ORF1a contains the sequences for NSPs 1-11 and ORF1ab for NSPs 1-16 [15,16].

Variations in the virus structures and essential enzymes are limited by their function, and there are regions in most proteins where a high degree of identity of the amino acid sequences or at least structures can be found. Only the receptor-binding region of the S-protein seems to be exclusive to each strain, although some functional constraints are apparent.

Findings on SARS-CoV-2 linear epitopes have already been included in early stage bioinformatics’ prediction of human B- and T-cell epitopes [17,18,19], phage display studies, peptide microarrays, and T cell activation assays [20,21,22,23,24]. However, the number of epitopes from each study or the resolution of the epitopes was limited. The method presented here enabled us to investigate the complete immune response by mapping all coronavirus antigens in parallel and to analyze the potential epitopes at the level of amino acid variations, for example, the naïve sequence and accepted variations. This is necessary because, due to the similarity of the antigens, it was expected and confirmed that healthy persons can have an established cross-reactive immune response to SARS-CoV-2 at B- and T-cell levels [25,26,27,28,29,30,31,32], although this is not always activated by vaccination [33]. This immune response is not restricted to the virus coat proteins but the T-cell response is also found for the NSP12 [34]. In addition, an existing immune response to other coronaviruses has in many patients a positive effect on the prognosis [35,36]. Multiple questions arise, particularly in cases in which the existing immune response can be protective or obstructive [37] and in which epitopes are actually responsible. 

This study aimed to broadly identify minimal epitopes from patient sera that are common or specific for the different coronavirus strains and are recognized by a sufficiently large number of patients, so that they can be used further in the characterization of patients. Despite the fact that HLA types have a significant influence on the recognition of individual epitopes [38], this was achieved with minimal quantities from a large number of different patient samples using a statistical method for epitope motif identification. This was possible with minimal quantities from a large number of different patient samples using a statistical method for epitope motif identification. Since epitope fingerprinting is based on statistical analyses of NGS data, it is possible to search for any additional antigen sequences for potential motifs in the data sets. Even antigens that are initially not of interest or additional epitope variants can be searched without depending on additional measurements using aged sera.

## 2. Materials and Methods

### 2.1. Patients and Resources

A total of 428 sera were obtained from the central laboratory of the Klinikum St. Georg (Ethics committee registry number EK-allg-37/10–1). Sera were obtained from patients with known COVID-19 infection based on PCR-testing or individuals undergoing vaccination with the vaccines BNT162b2 mRNA (BioNTech), mRNA-1273 (Moderna) and AZD122 (AstraZeneca). Patients were recruited upon admission or during the clinical course or follow-up based on written informed consent. In addition, we used 105 sera from Invent Diagnostics (Ethics committee registry number/feki code 011/1763) (respiratory disease) and a Biobank of sera from food allergen sensitized subjects at Fraunhofer IZI (Ethics committee registry number 202/16-ek).

Epitope Fingerprinting (see also Appendix A):

A special naïve peptide phage display was used in the selection experiments [39], and only two selection rounds were applied, as previously described [1]. Selections from the ENTE-1 peptide phage display library were performed using Dynabeads Protein A (Thermo Fisher Scientific Inc., Waltham, MA, USA). For each sample, 50 µL beads were added to 20 µL serum in 100 µL of 0.1% *v*/*v* Tween^®^ 20 in PBS (pH = 7.6) for one hour and then washed twice with wash buffer (0.1% *v*/*v* Tween^®^ 20 in PBS, pH = 7.4). After washing, the beads were resuspended in 200 µL PBS (pH7.4). Two rounds of selection were conducted. A total of 100 µL coupled beads were incubated for 2 h with 4.0 × 10^11^ cfu (with respect to 1.000-fold, the cfu output from the first selection round was used) in 1 mL of wash buffer containing 2% *w*/*v* BSA. Samples were washed 5× with 1 mL of 0.1% *v*/*v* Tween^®^ 20 in PBS for the first round and 5× with 1 mL of 0.5% *v*/*v* Tween^®^ 20 in PBS for the second round. The washed beads with bound phage particles were added to a bacterial culture. Phage rescues have been described. Pooled DNA of the recovered phagemids from the first and second rounds of selection was subjected to NGS in an Illumina MiSeq, as previously described. Oversampling and, thus, excessive data collection was not necessary, since in this case, the library design based on trinucleotide synthesis provides a strict framework of allowed nucleotides. This allows the detection and removal of sequences with potential sequencing errors after low-quality sequences were removed and the back and forward runs were combined using PEAR and processed using Trimmomatic (EMBOSS software package [40]). Finally, data sets from each sequencing run were cured from sequencing errors and other artifacts, as described in the latest version of the LibDB software (epitopic GmbH, Leipzig, Germany) [1,41]. Any sequences deviating from the library codon structure were sorted out in this procedure because they potentially contained additional reading errors by the sequencer. This resulted in a loss of up to 40% of sequences, which is expected with respect to the reading length and published reliability data for the sequencer. This allowed us to search for the statistical enrichment of amino acid 3-mers, 4-mers, and 5-mers with one variable position matching the antigen sequence. In this case, data sets from all selection rounds comprising 180,000 to 350,000 16-mer sequences of the library’s variable regions were subjected to the analyses.

In the first step for each data set, the statistics of all motifs in the data sets were calculated. The statistical value of occurrence was calculated versus the amino acids expected from the design of the library. Since the starting library has reproducible and predictable statistical distribution of the amino acids in each position of a 16-mer random sequence, it can be expected that any enrichment is caused by the selection experiment. Special software and the MySQL database allow for the retrieval of all peptide sequences containing a specific motif, and further analysis by the alignment can reveal potential similarities beyond a central motif. Finally, peptides were selected based on the individual alignment results from both the naïve sequence and strongly enriched phage-displayed motifs, particularly when several variants were surrounded by two cysteines or other conserved amino acids.

### 2.2. Peptide Arrays

Peptides were purchased from peptides&elephants GmbH (Hennigsdorf, Germany). All peptides had a C-terminal ebes-ε-azido-Lys linker, so they could be printed and immobilized as triplicates on DBCO-coated glass slides using click chemistry [1]. The surfaces of these slides was prepared via silanisation. A stock solution of 50 mg/mL DBCO-amine in DMF (molecular sieve dried) was prepared for this purpose. The slides were incubated with a 120 µL DBCO solution (0.25 mg/mL) and incubated overnight in the dark at room temperature. They were then washed with EtOh and centrifuged dry. Slides were stored at −20 °C. 

Each slide was blocked for one hour at 4 °C in an array-buffer (PBS containing 0.1% *v*/*v* Tween^®^ 20 and 1% *w*/*v* Casein, pH 7.4), which was also used for all further solutions. The slides were incubated for 2 h at RT with the patient serum diluted 1:50, washed twice (PBS containing 0.1% *v*/*v* Tween^®^ 20, pH 7.4), incubated for one hour at RT in a solution of mouse anti-human IgG antibody (1:5000; Abcam, Cambridge, UK), and washed twice again. The antibody binding was detected by incubating the array for one hour at RT with a secondary Cy5-labelled goat anti-mouse antibody (1:5000; ThermoFisher Scientific). Finally, the slides were washed again twice and the fluorescens was measured in a microarray reader at 10 μm resolution using a laser at 532 nm with 25% power/PMT Gain 600 and 635 nm with 25% power/PMT Gain 600 (Genepix 4300; Molecular Devices, San Jose, CA, USA).

Determination of spot intensities (see also Appendix A):

The first step was the detection of the a priori-known grid (GAL file) in the image, as the grid and spots may be rotated or shifted. Therefore, we used an image correlation technique that observes the whole grid at once to estimate the exact location of the grid. Afterwards, the location of each block undergoes another, more precise, correction to address the block specific rotation and translation. This step was corrected if necessary and validated before continuing with the analysis.

With the known position of each spot, we used a segmentation approach that combines a seeding threshold and a masking threshold with a geodesic dilation to distinguish between the fore- and background signal. This shall result in one or a few connected segments that surround the exact shape of the spot. To address single pixel inaccuracies, the shape becomes blurred with morphological binary operations.

In the last step of image analysis, we extracted information regarding the fluorescence intensity and shape of the segmentation. Information about the shape of a segment was mainly used to filter invalid segments due to artifacts.

As a reliable and robust method to decide a positive or negative spot result, we used the total count of all pixels in the spot after subtracting the background per pixel, summarizing all intensities in a spot segment and subtracting the median of the block background for every pixel.

The raw total fluorescence signal intensity of triplicates was used to calculate the upper/lower quartile. The difference of the upper and lower quartile was multiplied with 1.5. These number was added to the upper quartile and subtracted from the lower quartile. All raw fluorescence signal intensity not located in this calculated range was removed. The adjusted raw fluorescence signal intensity was calculated as a multiple of the background (water). 

## 3. Results

### 3.1. Epitope Fingerprinting

Data from selection experiments with more than 200 sera samples from COVID-19 and vaccinated patients were matched to structural and non-structural coronavirus proteins. In contrast to standard epitope mapping, this bottom-up approach identifies a variety of sequences similar to the antigen potentially recognized by the antibody. Statistics are calculated on the basis of the library design (see Appendix A for further information).

Instead of trying to obtain all potential epitopes from all patients, we focused on the minimal peptide epitope sequences commonly found when searching for data from COVID-19 serum selection experiments. This included potential epitopes of either SARS-CoV-2 or the four endemic coronavirus strains (eCoV) HKU1, OC43, 229E, and NL63. We were able to identify more than 100 epitopes by mapping all structural proteins (S-, N-, M-, and E-proteins) as well as several other proteins of all five strains (see Table 1) by applying in silico analyses to data from sera, including 80 COVID-19, 28 vaccine, and 11 respiratory disease sera collected in 2018. As the focus was on broadly recognized and short peptide epitopes, many more candidates were not used here because they lacked matching data in different patient sera.

Statistical evidence for epitopes corresponding not only to SARS-CoV2 proteins but also from other coronaviruses (HKU1, OC43, NL63, and 229E) was found in many sera, even in patients with COVID-19. Potential epitope-related peptides were chosen either from the naïve antigen sequence considering the available protein structures, that is, the S-protein, as well as alternative conserved amino acids in phage-displayed sequences. They contain enriched antigen motifs surrounded by different N- and C-terminal sequences, which often generate constrained or otherwise diverse structures. The goal was to select an appropriate minimal peptide sequence based on the virus protein as well as peptides that were directly derived from the enriched phage sequences, particularly disulfide-circularized peptides. This allows for the identification of peptide mimotopes recognized by a variety of patient-specific antibodies that bind to the same immunogenic region. This was difficult to achieve for the RBD region of the SARS-CoV2 S-protein, because the immune response was heterogeneous, whereas more common epitopes were found for the C-terminal domain. An alignment for all peptide epitopes with the virus proteins of all strains is shown in the Appendix A.

Figure 1 shows a simplified alignment of sequences retrieved from typical NGS data sets for peptides from sera. Enriched sequences with overall enriched 4-mer motifs in different data sets for the region preceding and following the conserved second furin cleavage site of the S-protein were retrieved and aligned. Already, the alignments show that different sera are likely to contain antibodies against different parts of this region. The results obtained from our approach also show the dynamics in the paratope visible in the alignment of the selected phage-displayed sequences, which led to the selection of individual peptides . For example: 

Some of the sera from the respiratory disease group (label “I”) bind to the KPSK motif, although it is not present in the antigens to which these patients have been exposed. Therefore, sequences from the eCoV sequences were selected (2-S-815Na1, H-S-815Na1, and O-S-815Na1).

KPSK is often followed by another lysine instead of the arginine in the SARS-CoV2 sequence; there is little overlap between the recognition of the sequences preceding and following the cleavage site, but there are exceptions. 

It is likely that antibodies recognizing the uncleaved protein are only present in a few sera samples, and in others, they are at least not dominant. Therefore, the C-terminal C-S-815Na2 and the “full size” C-S-815Na1 with a length corresponding to the aligned sequences’ overlaps were chosen for the peptide array.

### 3.2. Validation of Peptide Epitopes in Peptide Arrays

All epitope candidate peptides were synthesized and immobilized using click chemistry through a C-terminal linker on glass slides, considering that, at least for mimotopes, the free N-terminus should resemble the presentation on the phage particle. Here, we summarize the results for peptides capable of binding IgG from at least one group of sera in the initial test experiments. To select the most useful epitopes, they were initially screened with 20–30 sera. The peptide arrays were evaluated using novel software because spots generated by printing azide-coupled peptides on a DBCO-activated surface often vary in shape and intensity distribution, and the background of individual sera is highly different in intensity. The signal strength was calculated in comparison to the spots in the arrays generated by spotting water. There was little variance in epitopes selected as positive at two different thresholds, which is a good indicator that the array quality and software provide reliable results.

Four groups of sera were used for validation. Sera from patients diagnosed with COVID-19, a large group of sera from patients suffering from respiratory disease collected in 2018, sera obtained from vaccinated persons after the completion of the vaccination with mRNA-1273 (Moderna), tozinameran (Pfizer/BioNTech), and AZD122 (AstraZeneca) and a control group selected from a previous study on food allergies showed at least some low-level statistics for potential coronavirus epitopes according to the in silico epitope fingerprinting of existing data sets. However, the latter was not confirmed in the array measurements.

Figure 2 presents a general overview of IgG binding for 47 peptides selected from 167 initially tested peptides binding IgG from different sera. Figure 3 shows a summary of the array results. A broad variety of peptide epitopes have been identified, including one peptide identified in ANCA patient sera positive for proteinase 3 anti-neutrophil cytoplasmic antibodies (PR3-ANCA) [42]. The thresholds for the measurements were set to 20 with a five-fold background, without major changes with respect to the specificity of the selected peptides.

According to the results (Figure 3), antibodies to the peptide epitopes can be classified as SARS-CoV-2 specific epitopes, cross-reactive epitopes occurring after vaccination, or broadly recognized epitopes, which can be found particularly in patients with respiratory disease. The potential cross-reactivity of epitopes was initially estimated from the sequence similarities between different viral strains and the amino acids identified as relevant for binding as identified by epitope fingerprinting. Cross-reactive epitopes are found in structural proteins, such as RNA polymerase, as well as structurally conserved parts of the S-protein. Based on the results with sera, we identified further cross-reactivity, most likely based on structural similarities. 

The COVID-19 sera used in this study were obtained from patients after hospitalization. The number of days after the first symptoms, following the patient number in Figure 2, does not indicate the number of days after infection. However, it can be observed that the antibodies’ overall signals do not change or decrease even after 180 days or more. This is essentially true for any epitope, although in addition to the loss of expression, there may be a shift in specificity with the ongoing maturation of IgG and, therefore, changed affinity to the peptide after 6 months or more. The intensities for different measurements vary by 2–3 orders of magnitude by epitope and patient. Therefore, Figure 2 presents the data by threshold level. A heatmap based on the measured intensities can be found in the Appendix A.

### 3.3. Epitopes of Proteins from eCoV Strains

A large set of sera collected from patients with respiratory disease allowed the validation of a set of peptide epitopes that correspond primarily to HKU1, 229E, or OC43 spike protein sequences (peptides coded S-815). In particular, antibodies to this furin cleavage site have been found in a large number of sera. This site is of special interest because the partially identical sequence in the SARS-CoV-2 protein is recognized by antibodies as well. Remarkably, the pre-COVID-19 patients’ IgG showed no cross-reactivity with the selected SARS-CoV-2 sequences. On the other hand, there is a strong cross-reactivity of these patients’ IgG with sequences from endemic strains. Some of the peptides selected present the processed S-protein after furin cleavage than the intact protein sequence, since the statistical data suggested that some antibodies only recognize the novel N- and C-terminal sides of the cleaved proteins. Minimized peptides composed of the essential amino acids selected through the statistical approach have a higher selectivity than recently published peptides used to identify former OC43 infections among COVID-19 patients [35].

A special case of a structural S-protein epitope is the 2-S-614 epitope (614-NVRCVELL) identified in some COVID-19 sera. Two mimotope sequences were derived from phage peptides enriched with sera from vaccinated and infected individuals. A similar enrichment for sequence motifs was found from COVID-19 sera covering the sequence 741-YICGDSTECSNLLLQYGSFCT, but the statistical data were not consistent enough to encourage peptide synthesis, but the first tests with a SARS-CoV-2-related mimotope are now successful and the epitope was recently described [43]. This is a structurally conserved motif in the second domain of the S-protein, with the structure maintained by an identical pattern of disulfide bridges (Figure 4).

For the M-protein, the epitopes C-M-152Na1, C-M-15Ph1, and C-M-15Ph2 seem to be specific for SARS-CoV-2, but with low titers and apparent cross-reactivity with the vaccine sera. N-terminal epitopes C-M-8Na1 and H-M-5Na1 for the N-terminus do not share sequence homology, but are recognized by many sera. In addition, the OC43 derived E-protein epitope O-E-71Na1 is unique in sequence, but is primarily recognized by sera from vaccinated or infected patients. Since there is no M- or E-protein antigen in S-protein-based vaccines, we have no clues about which epitope structures these apparent mimotopes from eCoV strains exactly represent. Undoubtedly, binding antibodies are indicators of COVID-19 or the most recent coronavirus infections.

### 3.4. SARS-CoV-2 Specific Epitopes

Several short peptide epitope candidates have been identified in the receptor-binding domain of the S-protein in individual sera. C-S-350, C-S-431, and C-S-448 represent those recognized by IgG in the largest number of patients tested. Similar antibodies were also found in the vaccine sera. 

Among the potential N-protein epitopes, only C-N-376Na1 appeared to be useful in this broad array test. Another epitope, 53-NTASWFTAL-61, was found to bind IgG in vaccine sera. Investigation of potential cross-reactivity showed an enrichment of peptides corresponding to the N-terminal sequence of S-protein 60-SNVTWFHAIHVS-71 in the vaccine sera, which was much stronger than that in the sera of patients. Because the peptide statistics in patient sera allow both variants of a WFxA(I/L) motif, antibodies to both epitopes are likely in the patients’ sera and, of course, only to the S-protein in the vaccine sera (Figure 5). This region is affected by mutations in the alpha and omicron (B.1.1.529) lineages (A67V, Δ69–70), which makes it less similar to the N-protein. Data sets from the most recent patients did not show enrichment of S-protein motifs in this area, but they still showed antibody binding to the peptide in the array.

For the furin cleavage site in the S-protein, statistical data analyses predicted groups of antibodies binding to the full or either side of the cleavage site (Figure 1). All peptide epitopes matching this region showed the highest signal for all peptides tested in the array (see Appendix A). The peptide covering most of the sequence preceding this site (C-S-815Na1) is recognized less frequently by the vaccine group than by the COVID-19 patients, whereas the peptide C-S-815Na2 and the highly homologous H-S-815Na1, omitting several of the positively charged residues, are recognized with high frequency by sera from both groups. Both HKU1- and 229E-derived sequences actually seem to present a pan-coronavirus epitope recognized by all sera from patients with COVID-19 or another coronavirus infection, covering a highly conserved helical structure (Figure 4).

Since individual patient epitope patterns depend on HLA [38] as well as earlier infections by eCoVs, we did not expect to find a single peptide representative of all patients with COVID-19 or related to immunization. In addition, the subgroups investigated showed a high degree of potential previous HKU1 infection, dominating over the other eCoV as judged by the frequency of antibodies against specific epitopes.

### 3.5. Other Epitopes

Antibodies against other proteins, particularly RNA-polymerase proteins, were observed in all strains. The strongest binding was observed for IgG in COVID-19 patients and, to some degree, even in vaccine sera. Due to their conserved structures, these IgGs are likely to bind even slightly variant sequences in the non-structural proteins (NSP) of other coronavirus strains. This is expected, but little attention has been paid to the details of these epitopes, and limited reference data are available. It is worth noting that some vaccine sera have IgG binding to these peptides, although overall only weakly, indicating recent coronavirus infections.

One peptide epitope of the human proteinase 3, identified in patients with ANCA [41,43,44], was added to the array. PR-390-Na1 was identified in data from the sera of proteinase 3 positive ANCA patients. Initial data from severe COVID-19 showed details similar to ANCA, and indeed the peptide PR-390-Na1 bound IgG exclusively from several COVID-19 patients.

### 3.6. Comparison to Epitopes Listed in the Immune Epitope Database [44] (IEDB)

The number of epitopes listed for the four eCoV strains was limited when we began our project in 2020. Besides the furin II cleavage site epitopes, none of those validated in our study had been listed. However, presently, epitope data for these viruses and SARS-CoV-2 cover almost the entire S- and N-protein sequence. This might fit well with our observation that a very large range of individual epitopes could be expected, but this is surprising because of the broad coverage.

## 4. Discussion

Our data suggest a diagnostic advantage of using short, highly specific peptide epitopes. This enabled us to compare the individual immune responses to SARS-CoV-2, eCoV, and following vaccination. These peptides rely on analysis binders from a naïve library, including all types of variations in length and structural constraints, and are thus superior to simple linear fragment libraries.

Using a statistical phage display-based approach, we have been searching for short distinct peptide epitopes capable of binding IgG from a large number of patient sera. It is important to note that the reliability of the library is sufficient to allow calculations based on the library design and not on the overall statistics from the reference NGS data sets. In contrast to standard epitope mapping with peptides predefined by the antigen sequence [22,25,45,46,47,48] or statistical approaches based on less controlled peptide phage libraries [23,24,49], our approach allows for the identification of residues probably recognized by individual patients. Therefore, the selection of the peptide sequence can be confined to the essential conserved residues or residue types recognized by many different patients. Normally, such data are available by only using combined Ala-scan and peptide array approaches. The selected phage-presented amino acids surrounding the epitope motif can guide the use of structurally constrained peptides instead of the linear antigen sequence, which has been successful in several cases (for example, 2-S-614Ph4). Nevertheless, there are a variety of B-cell clones, and our data may only capture the epitopes of the most significant antibodies. On the other hand, data sets obtained from a single selection experiment on fresh serum can be used to search for multiple proteins, which we proved to be successful in this extended study.

The mapping of the COVID-19 B-cell response using peptide phage display methods has been described. Fragment libraries naturally restrict the results to naïve sequences and are comparable to peptide arrays, whether generated by random fragmentation [50] or selective synthesis [48,51]. Compared to the efforts to generate a well-characterized fragment library, peptide arrays are probably more cost-effective when applying recent technologies. Other publications are using random peptide libraries [51,52]. However, these methods are of limited use in statistical analyses because of the limitations in the composition of the applied libraries or the small number of enriched peptides. The library used here enables the identification of subtle variations due to the even (i.e., statistically predictable) amino acid distribution in the random sequence. These can be used to select peptides with selectivity without further modifications rather than shortening the peptide sequence. The natural limits of this approach are similar epitope motifs, for example, C-N-376Na1 or C-N-53, which confirms us in our approach that all peptide epitopes for the binding of serum antibodies must be tested in the end.

A set of peptides recognized by IgG of most or at least many patients was identified at a success rate above 50% per identified antigen sequence area. Some peptides are only recognized by a few patient sera but are still not recognized by the control sera. Other peptides are hydrophobic and result in signals from the majority of the sera investigated (for example, H-M-5Na1). However, a comparison with control sera shows that even this peptide preferably binds antibodies of infected or vaccinated patients, although it was initially attributed to HKU1. Applying this bottom-up method with sera from COVID-19 patients to the main proteins of all coronavirus strains allowed the screening of many more proteins than is probably feasible with peptide arrays. Due to individual variations in epitope recognition, it is necessary to combine at least two peptides for a sufficiently specific test. We have listed such tables in the Appendix A (Excel data file EpitopeSelectivity). Several combinations reach >90% specificity and sensitivity in the groups of patients studied. Since a large number of cross-reactive epitopes from eCoV seem to be recognized by both vaccinated and infected persons, and there is an apparently high background of recent HKU1 infections, microarray tests should be repeated with different patient populations and more control sera. This could identify universal epitope biomarkers for infections caused by different eCoVs and SARS-CoV-2.

However, it also generates unexpected results, such as cross-reactive epitopes and at least two sequences specific for diseased and vaccinated patients, but apparently linked primarily to the immune reaction to the S-protein (N-N-71Na1) or general coronavirus infections (H-M-5Na1). Restricting the screening to a limited set of peptide epitopes in microarrays allowed for the efficient screening of a large number of sera, despite the limited sample volumes available in many cases. 

Several of our findings with respect to the observed cross-reactivities were surprising due to relatively large differences in the virus protein sequences, but were not completely unexpected due to the structural homologies of different coronavirus strain proteins (2-S-614). This is also because the amino acids surrounding the phage-displayed motifs of naïve libraries can support folds related to the target protein better than simple linear naïve sequences. 

We identified three different types of cross-reactivity:-Cross-reactivity based on the sequence;-Between coronavirus strains, as expected;-Between different SARS-CoV2 epitopes, as not expected at all;-Structural epitopes defined by conformation.

In particular, the apparently broadly recognized epitopes of the furin cleavage site S2′ in the S-protein, which for functional reasons is structurally and by sequence well conserved, showed an unexpected strain and patient status specific pattern. This region is called a protective [52] or universal T-cell epitope [28], and antibodies targeting this site have been studied in detail [53]. According to our results, this is apparently more complex and depends on whether the B-cell immune response has been raised against SARS-CoV-2 or another coronavirus, and which amino acids were recognized by the immune system in this case. Sequence statistics from different patients obtained in this study indicated at least two types of antibodies. They recognize the entire sequence surrounding the cleavage site; for example, the majority of antibodies after vaccination, or they preferably bind either the N- or C-terminal parts of the processed protein (Figure 1). Therefore, different types of peptides (all labeled ‘S-815’) have been tested. General cross-reactivity between different coronavirus strains at the peptide level has been described [48] when using a peptide covering the entire region from OC43 and SARS-CoV-2. Yamagochi [35] observed substantial cross-reactivity, which apparently protected some patients from severe diseases. Using almost the same but sequence-optimized OC43 peptides, cross-reactivity was confirmed. However, with shorter peptides covering only parts of this cleavage site for SARS-CoV2, only a small group of respiratory disease patients showed cross-reactivity with SARS-CoV-2 epitopes. In contrast, the majority of COVID-19 patients’ antibodies can recognize epitopes corresponding to the HKU1, OC43, or 229E sequences. This discrepancy is striking, since these peptide epitopes differ only by a few residues, which could be considered minor variations. The origin of these cross-reactive antibodies could be a small part of the memory B cell pool originating from an earlier infection with coronaviruses. Only a minority of cross-reactive clones are boosted in COVID-19, but these are probably barely measurable at the antibody level in HKU1-infected patients. This is a good explanation for the absence of CoV-2-neutralizing activities in older sera [54] and the beneficial effects of earlier eCoV infections [26,35,54,55].

A second unexpected result was that vaccine-induced antibodies showed little or no binding to the N-terminal peptide (C-S-815Na1), but almost all showed binding to the peptide comprising the C-terminal part (C-S-815Na2), whereas COVID-19 patients had an IgG against both. In infected patients, the furin-cleaved S-protein is the major protein variant present in the immune system, which might explain this bias. Simultaneously, it may be speculated that antibodies binding to the processed protein in infected patients could still prevent the fusion of the virus with the cell. With respect to the application of mRNA vaccines, this could mean that the absence of furin expression in skeletal muscle [56,57] could have caused differences in the immune response.

Structural cross-reactivity was observed for the loop of a four-helix bundle in the S-protein. Cysteine-constrained peptides identified directly from phage clones, e.g., 2-S-614Ph2 (Figure 4), with a part of the 229E sequence of one loop that was selected due to the enrichment with patient sera, are recognized by antibodies against the CoV-2 protein. Antibodies from respiratory disease patients identified the peptides as 229E-specific within this group, but the data with sera from vaccinated persons revealed a high degree of cross-reactivity. Using the SARS-CoV-2 S-protein in the peptide database search did not result in statistical data, encouraging the synthesis of individual peptides. When multiple cross-reactive antibodies are present in serum, the approach reaches its limits in silico and requires peptide testing. The most important finding is that despite the limited sequence identity, the homology of the structures caused by the conserved pattern of Cys is sufficient for cross-reactivity; therefore, an applied library with potentially constrained cysteines is optimal for identifying mimotopes for antibodies against structurally constrained epitopes, which are usually not accessible with synthetic peptides, unless special efforts are made for structural motifs [43].

We found additional cross-reactive epitopes in the structural proteins of several strains. An existing immune response to eCoV proteins is observed in practically all sera of patients with COVID-19, and antibodies can often be detected in healthy individuals. It can be assumed that a stable set of memory B cells originating from earlier eCoV infections persists for at least several years. This is an important aspect in estimating the protective effects of infections and vaccinations. We investigated the development of antibody subclasses for different epitopes, which are apparently far more complex than the data presented herein.

Even intra-proteome cross-reactivity could be found for an epitope initially attributed to a sequence in the SARS-CoV2 N-protein. Our tentative conclusion is that this is a case of cross-reactivity between two related sequences in the same viral proteome. The immune system of COVID-19 patients most likely generates a variety of antibodies against similar N- and S-protein epitopes. This could be the reason why the IgG-binding signal from vaccine sera is constantly high against a single antigen target, while it varies substantially between different COVID-19 patient sera. Whether this is a coincidence or actually helps the virus escape the immune response in infections remains to be investigated. The presence of a strong antibody signal against this epitope could be a reliable indicator of a recent vaccination.

The large number of epitopes recognized in non-structural proteins allows for the generation of more general diagnostic tools for coronaviruses. These antibodies have not been studied routinely thus far, and only recently has T-cell cross-reactivity been described [34]. They do not contribute directly to the protection against the virus by antibodies, but, as can be observed from the general frequency in all sera, their levels will allow differentiation against recent infections, even in vaccinated individuals. 

## 5. Conclusions

Epitope identification is an efficient way to identify individual patients and antigen-specific variations in the immune response. The peptide epitope sequences identified here have the potential to enable the stratification of patients for vaccination and previous or ongoing infections with SARS-CoV-2 and eCoV. The data sets of sequences will allow future searches for antibody epitopes not yet in our focus. Moreover, since certain peptides could be expected to stimulate specific memory B cells, the stimulation of a selective immune response may in the future depend on such selective boosters instead of full-protein vaccines. The results may also inspire the future design of (booster) vaccines for the ongoing pandemic, and this approach could be applied in the development of other vaccines as well.

## Figures and Tables

**Figure 1 vaccines-11-01403-f001:**
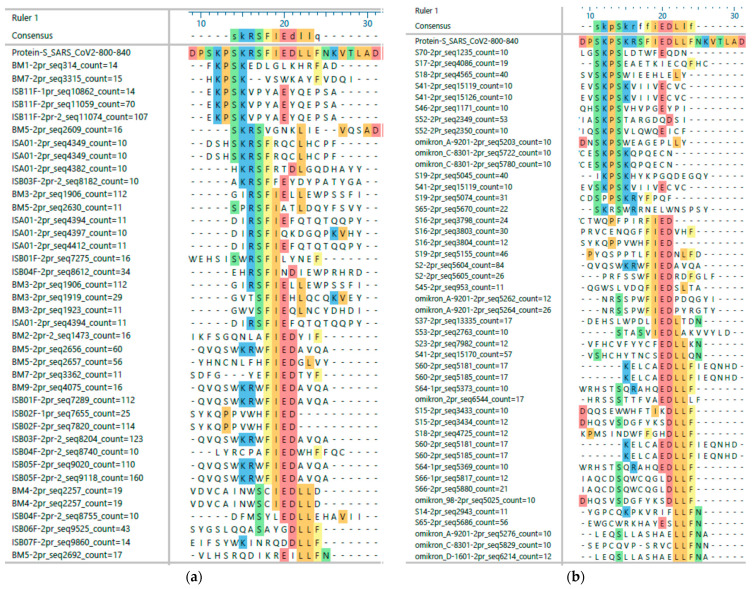
Alignments from sequences enriched for the second furin cleavage site epitopes in the S-protein of SARS-CoV2 showing different antibody recognition patterns between vaccination (**a**) and COVID-19 (**b**). For this example, all sequences with 4-mers identical to the sequence KPSKRSFIEDLLFNK were retrieved from 78 data sets of COVID-19 and 29 data sets of vaccinated patients, resulting in 6790/11,083 different sequences found 13,256/22,826 times in 46 Mio. and 18 Mio. sequences in total. The alignment shows only sequences found at least ten times with a 5-amino-acid identity to the antigen. Amino acids identical with the antigen are highlighted by colors indicating their chemistry, the antigen reference sequence is fully colored. The data set and sequence IDs are internal patient numbers: first (1pr) or second (2pr) selection rounds, and sequence number and frequency ‘count’ found in the NGS data set. The first and second selection data sets often contained different data sets with occasionally identical sequences.

**Figure 2 vaccines-11-01403-f002:**
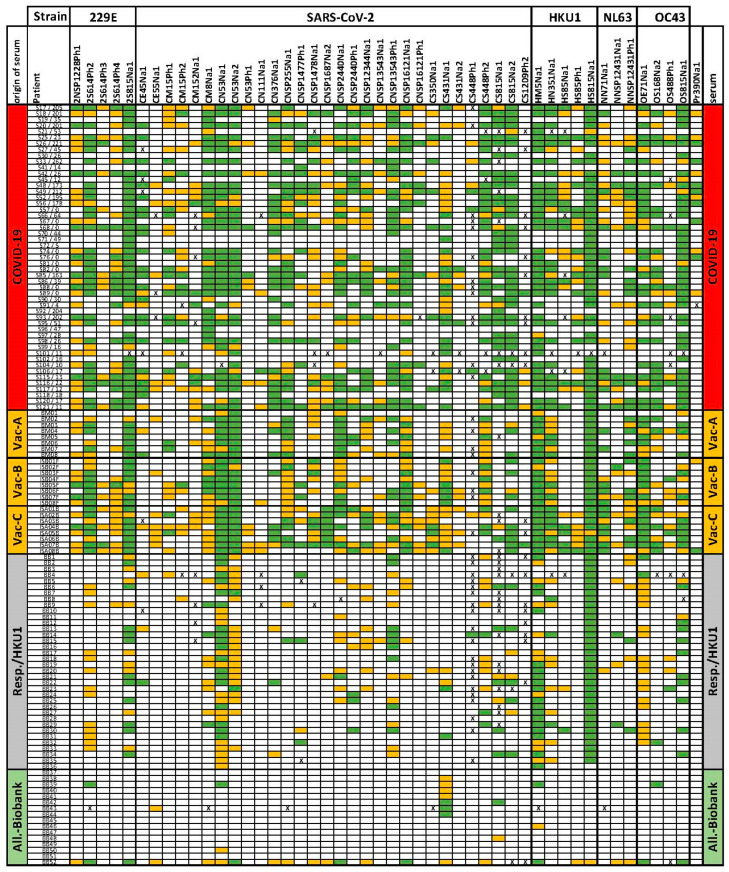
Overview of all observed IgG binding to epitope peptides (sequences shown in Figure 3). Applying a background cut off at 5× (yellow) and 20× (green), the background is the intensity of spots without peptides. COVID-19 patient numbers Sxxx are followed by dashes and days after the first symptoms (d.a.f.s.). The epitope codes are explained in Figure 3. Origin: patients = COVID-19; vaccinated individuals = Vacc (A = Moderna, B = Pfizer/BioNTech, and C = AstraZeneca); pre-COVID-19 respiratory disease patients with HKU1 epitopes = Resp./HKU1; allergy biobank sera with potential coronavirus antibodies = All.biobank. All numbers represent internal serum numbers. X: peptide array values below background or with too high a standard deviation of triplicates. A heatmap of the signal intensities is provided in the Appendix A.

**Figure 3 vaccines-11-01403-f003:**
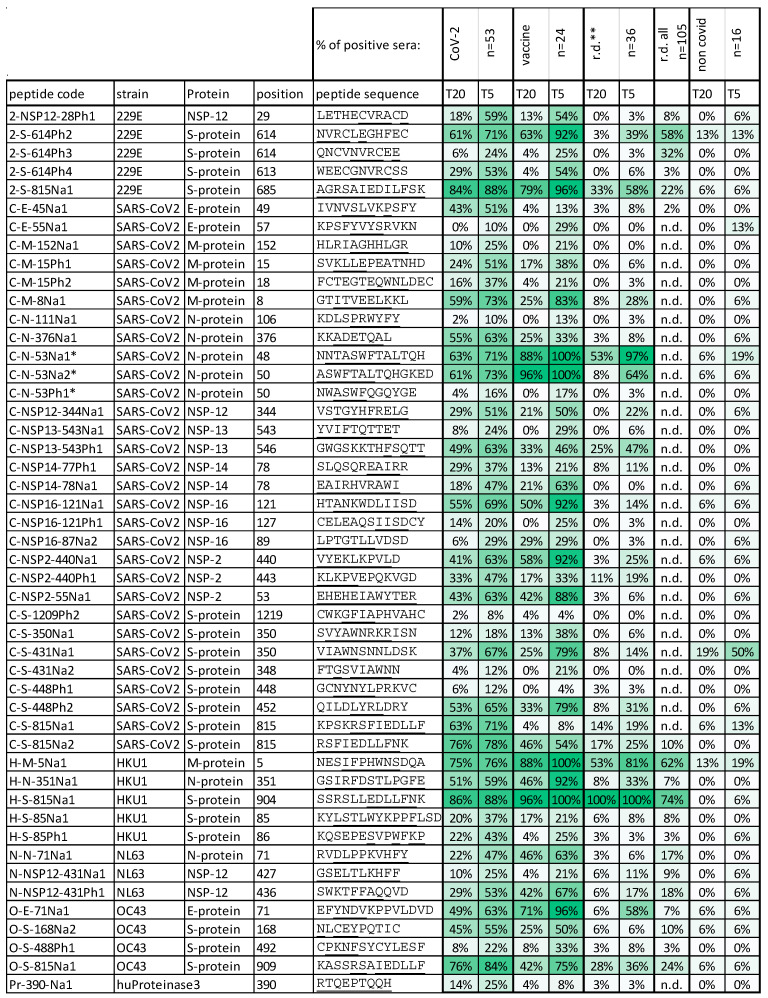
Epitopes, mimotopes, and serum reactivity. Peptide codes: 2/C/H/N/O, first letter of strain; S/N/E/M/NSPxx protein code; Number = first residue position in first discovered sequence; Na-naïve/Ph phage-derived sequences and consecutive numbers. The positions for identical and homologous regions were kept in the epitope code throughout the sequences of all strains. * C-N-53 cross-reacts with an S-protein epitope. Position: First amino acid of the underlined motif within the peptide as identified in sequence data (Table 1); patient data for 5× and 20× thresholds in array measurements, and labelled with green heatmap: COVID-19 patients; vaccinated (altogether); R.D. respiratory disease with R.D. ** potential HKU1 patients only; R.D. all: data from parallel testing with a different set of peptides specific for eCoVs (see Appendix A), not containing all epitopes; non-COVID-19: healthy control.

**Figure 4 vaccines-11-01403-f004:**
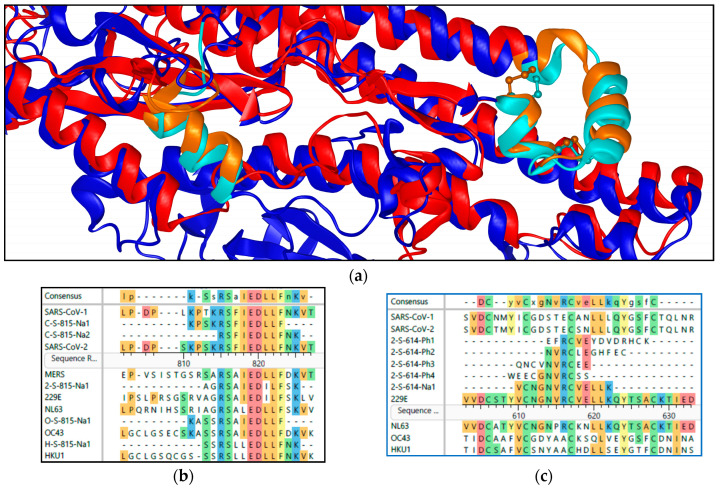
Structural identity even with different sequences: (**a**) Alignment of SARS-CoV-2 (red, pdb:6vxx) and 229E (blue, pdb:6u7h) structures of the second domain of the S-protein. In light colors, furin cleavage site 2 in the left center (epitopes C-S-815) and the cross-reactive loop including conserved cysteine side chains (epitope 2-S-614) on the right. (residues 229E: Cys606-Cys628 and 687–699, CoV-2: Cys720-Cys742 and 810–825). Below the sequence alignments of the same regions, the peptides used in the arrays are aligned to their parental sequences. (**b**) The peptides “815er” aligned are all binding antibodies in different patient sera. (**c**) For the loop motif, apparently only the Cys pattern is conserved. Only peptides 2-S-614-Ph2/3/4 exhibited binding. (Coloring as Figure 1).

**Figure 5 vaccines-11-01403-f005:**
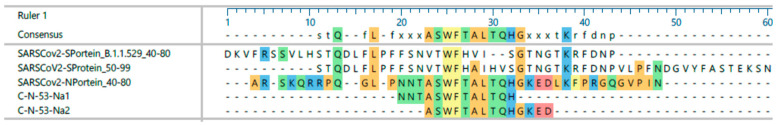
Alignment of cross-reactive peptides C-N-53Na1 and C-N-53Na2 with the N- and S-protein sequences from SARS-CoV2 and the omicron variant. (Coloring as Figure 1).

**Table 1 vaccines-11-01403-t001:** The sequences for the different proteins were taken from the following database entries.

	S-Protein	RNA-Pol	E-Protein	M-Protein	N-Protein	NSP
CoV 229E	AOG74783	AIW52761	AOG74785	AOG74786	AOG74787	AGT21366.1
CoV HUK1	AYN64561	ABD91892	AGW27883	AYN64564	AYN64565	YP_459942
CoV NL63	BBL54116	AIW52828	AFV53150	AFV53151	AFV53152	QII57165.1
CoV OC43	AMK59677	AIX10747	AMK59679	AMK59680	AMK59681	YP_009555257
SARS CoV	ABD73002	QJE50587	AAP13443	AAU07933	ABI96968	NP_828870
SARS CoV 2	QII57161	QND77388	QIH45055	QII57163	QIH45060	YP_009725308.1
SARS CoV 2 (omikron)	UJK14488 (B.1.1.529)					

## Data Availability

The data that support the sequence findings of this study are available from the corresponding author upon reasonable request. Array data intensities can be found in the Appendix A.

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
