# Peer review of "Detection of Antibodies against Endemic and SARS-CoV-2 Coronaviruses with Short Peptide Epitopes"

_vaccines, 2023, doi:10.3390/vaccines11091403_

Round 1

Reviewer 1 Report

The study by Szardenings et al. aimed at identifying minimum peptidic region in Coronaviruses both endemic seasonal (hCoV-229E, -NL63, -HKU1, and -OC43) and the other pathogenic strains including the SARS-CoV2, SARC-CoV and MERS. The notion of this exercise was to use the identified peptides that are specific for the different coronaviruses and apply them in characterization of the patient sera. Another expected outcome of the study was that the peptides identified could possibly be used in a diagnostic test to differentiate previous infections and COVID-19.  The results however presented does not satisfactorily suggest that the authors were able to achieve their goal due to the higher degree of cross-reactivity between the samples and the high percent similarity among antigens. There are some peptides that show differential binding to the sera from vaccinated vs infected individuals however, the differences are not very drastic. The article presentation and writing is hard to follow and it is unclear to me what the inferred outcome of this article is. There are several grammatical mistakes as well as errors in the sentences making it hard to follow-through.

The sentences are very hard to follow. The authors seem to get confused between peptide, sequence, structures. 

Author Response

General aspects are answered in the attached letter, because the questions and comments of the reviewers overlap to some degree. In this case we are missing real questions, but the overall comment on the statements given can be somehow summarized:

Reviewer 1

The study by Szardenings et al. aimed at identifying minimum peptidic region in Coronaviruses both endemic seasonal (hCoV-229E, -NL63, -HKU1, and -OC43) and the other pathogenic strains including the SARS-CoV2, SARC-CoV and MERS. The notion of this exercise was to use the identified peptides that are specific for the different coronaviruses and apply them in characterization of the patient sera. Another expected outcome of the study was that the peptides identified could possibly be used in a diagnostic test to differentiate previous infections and COVID-19.  The results however presented does not satisfactorily suggest that the authors were able to achieve their goal due to the higher degree of cross-reactivity between the samples and the high percent similarity among antigens. There are some peptides that show differential binding to the sera from vaccinated vs infected individuals however, the differences are not very drastic.

Answer: We have added tables describing more clearly the differences, selectivity and sensitivity of the peptides by the peptide pairs (additional supplementary file). This was shown already in figure 3, but we have now also added a heat map. > 90% sensitivity resp. selectivity for non-optimized peptides are reasonable in our understanding. We also added a very recent reference (38), describing the significant effects of different HLA groups and explained the point of the unlikelihood of single peptide immune diagnostics (line 376-378, 426-438)

The article presentation and writing is hard to follow and it is unclear to me what the inferred outcome of this article is. There are several grammatical mistakes as well as errors in the sentences making it hard to follow-through.

Answer: We optimized the writing, but putting all data into this article is indeed difficult and as we see above the reviewer also missed some points regarding the epitopes’ diagnostic value.

Reviewer 2 Report

The manuscript “Detection of antibodies against endemic and SARS-CoV-2 corona viruses with short peptide epitopes” described peptide-based method to identify epitopes that can be recognized by serum from patients with COVID-19 or vaccinated by S-protein based vaccines. They have identified the cross-reactive epitopes between coronavirus strains, different SARS-CoV-2 epitopes. The paper is interesting and could be useful to develop high throughput diagnostic method for the epidemic and endemic coronavirus infections. A couple of questions should be addressed before the work is accepted for publishing.

Suggestions,

1.     Please provide more information on the peptide pool for peptide array analysis. Such as the length, the coverage of the viral proteins and the overlaps between adjacent peptides, and the sequence information of the peptides.

2.     Based on your analyses, could you conclude which peptides can be used to set up antibody detection assays for SARS-CoV-2?

3.     When discussing peptides that are suitable for diagnostic purposes, please provide the percentage of positive reacted samples.

Author Response

In addition to these answers see in general the attached letter and with respect to overlapping questions, our answers here are short:

Reviewer 2

The manuscript “Detection of antibodies against endemic and SARS-CoV-2 corona viruses with short peptide epitopes” described peptide-based method to identify epitopes that can be recognized by serum from patients with COVID-19 or vaccinated by S-protein based vaccines. They have identified the cross-reactive epitopes between coronavirus strains, different SARS-CoV-2 epitopes. The paper is interesting and could be useful to develop high throughput diagnostic method for the epidemic and endemic coronavirus infections. A couple of questions should be addressed before the work is accepted for publishing.

Suggestions,

  1. Please provide more information on the peptide pool for peptide array analysis. Such as the length, the coverage of the viral proteins and the overlaps between adjacent peptides, and the sequence information of the peptides. Answer: It is mentioned in the text, that we use a naïve library, which has been used previously (reference 1). We believe the reviewer is mistaken by related research with peptide/gene fragment libraries derived from the viral proteins (also cited in the text)

  1. Based on your analyses, could you conclude which peptides can be used to set up antibody detection assays for SARS-CoV-2?

Thank you for this consideration: Due to the heterogeneity of the patients’ immune response there is not a single peptide sufficient to set up an assay. But actually pairs of peptides would already be sufficient for diagnosis of COVID-19. We have added this data in the supplement. Although, our patient group is dominated by recent HKU1 infections and in other regions a different eCoV background may change the results. Therefore we have been carefully selecting not to make this statement. This has been added in the text (line 426-438)

Reviewer 3 Report

The manuscript entitled ‘Detection of antibodies against endemic and SARS-CoV-2 corona viruses with short peptide epitopes’ by Michael Szardenings and coauthors is promising for initial step of future vaccine research. However, as preliminary result the author supposed to propose future direction of the next research for vaccine production.

Overall, the manuscript is well written, and the bioinformatics seems sound.

I suggest the following minor changes:

To judge the quality of the data, it would be highly beneficial to have some kind of heat map of peptide/epitope array results.

In Figure 1, labelling Figure 1a and Figure 1b (boxes) are overlapping with the alignment.

Fig. S1-1 needs minor adjustments: remove the line under ‘Epitope’ and adjust the arrows a bit.

Reread and check line 131-132: (kern)?

Author Response

In addition to these answers see in general the attached letter and with respect to overlapping questions, our answers here are short:

Reviewer 3

The manuscript entitled ‘Detection of antibodies against endemic and SARS-CoV-2 corona viruses with short peptide epitopes’ by Michael Szardenings and coauthors is promising for initial step of future vaccine research. However, as preliminary result the author supposed to propose future direction of the next research for vaccine production.

Answer: The reviewer is correct, but we have chosen not to do so. The results regarding tissue specific protein processing have already been communicated by us after the first vaccine sera had been tested in 2021 and we met too much resistance by several journals to publish that. Anybody understanding the data would actually draw his own conclusions. 

Overall, the manuscript is well written, and the bioinformatics seems sound.

I suggest the following minor changes:

To judge the quality of the data, it would be highly beneficial to have some kind of heat map of peptide/epitope array results.

Answer: We have added the heatmap to the supplement (Figure S4) and explained in the text, that the heterogeneity of the signal intensities (as can be seen in the heatmap), makes it difficult to understand the overall relevance of individual epitopes.

In Figure 1, labelling Figure 1a and Figure 1b (boxes) are overlapping with the alignment.

Answer: Corrected, MS Word formatting issues

Fig. S1-1 needs minor adjustments: remove the line under ‘Epitope’ and adjust the arrows a bit.

Reread and check line 131-132: (kern)?

Answer: Thanks, corrected